# EndoNuke: Nuclei Detection Dataset for Estrogen and Progesterone Stained IHC Endometrium Scans

Anton Naumov [1], Egor Ushakov [1], Andrey Ivanov [1], Konstantin Midiber [2,3,4], Tatyana Khovanskaya [2], Alexandra Konyukova [2], Polina Vishnyakova [4,5], Sergei Nora [6], Liudmila Mikhaleva [2,3], Timur Fatkhudinov [2,4] and Evgeny Karpulevich [1,*]

1   Information Systems Department, Ivannikov Institute for System Programming, Russian Academy of Sciences (ISP RAS), 109004 Moscow, Russia; vandedok@ispras.ru (A.N.); ushakov@ispras.ru (E.U.); ivanov.as@ispras.ru (A.I.)
2   A.P. Avtsyn Scientific Research Institute of Human Morphology, 117418 Moscow, Russia; midiberkonst@gmail.com (K.M.); zimavnebe@mail.ru (T.K.); have.to.study@yandex.ru (A.K.); mikhalevalm@yandex.ru (L.M.); tfat@yandex.ru (T.F.)
3   City Clinical Hospital No. 31 of the Moscow Healthcare Department, 119415 Moscow, Russia
4   Medical Institute, Peoples' Friendship University of Russia (RUDN University), 117198 Moscow, Russia; vpa2002@mail.ru
5   Laboratory of Regenerative Medicine, National Medical Research Center for Obstetrics, Gynecology and Perinatology Named after Academician V.I. Kulakov of Ministry of Healthcare of Russian Federation, 117997 Moscow, Russia
6   Yaroslav-the-Wise Novgorod State University, Department of Microbiology, Immunology and Infectious Diseases, 173003 Veliky Novgorod, Russia; sergey.nora@novsu.ru
*   Correspondence: karpulevich@ispras.ru

**Abstract:** We present EndoNuke, an open dataset consisting of tiles from endometrium immunohistochemistry slides with the nuclei annotated as keypoints. Several experts with various experience have annotated the dataset. Apart from gathering the data and creating the annotation, we have performed an agreement study and analyzed the distribution of nuclei staining intensity.

**Dataset:** http://endonuke.ispras.ru (accessed on 20 April 2022).

**Dataset License:** CC BY 4.0

**Keywords:** digital pathology; endometrium; nuclei detection dataset; whole-slide imaging

## 1. Introduction

One of the key problems in medical research and medical decision-making is the subjective assessment of clinical data, especially of a visual nature. In addition, an integrated approach to the assessment of biomedical data is often required: it is necessary to compare the results of several studies, and, as their number grows, the complexity of this process increases. Both problems can be solved by creating software tools that play a supporting role in making a medical or clinical decision. One of the areas where such instruments are especially needed are histological examinations, which are often required in patients diagnosed with infertility. Infertility is a disease of the reproductive system (ICD-10: N97), defined as the inability to achieve a clinical pregnancy after 12 months or more of regular unprotected intercourse. Infertility affects a large proportion of humanity, affecting over 186 million couples worldwide [1].

The receptivity of the endometrium is one of the critical factors affecting pregnancy chance. As the regulation of morphological and functional changes in the endometrium is caused by the effect of steroid hormones, the study of the number and ratio of high-grade progesterone and estrogen receptors in the stroma and endometrial glands is of particular interest [2].

Estrogenic activity in the endometrium is primarily carried out through the binding of its related nuclear receptors, namely estrogen receptor alpha (ER-$\alpha$) and estrogen receptor beta (ER-$\beta$). Progesterone receptors are also expressed in the nuclei of endometrial cells in the form of two functionally different isoforms: PR-A and PR-B. Evaluation of the immunohistochemical reaction is performed separately in the glands and stroma of the endometrium. The endometrium is not only the subject of histopathological diagnosis, but also an extremely valuable tissue for experimental manipulation and modeling of reproductive diseases using human biomaterial [3]. For instance, the endometrium can be used as a control tissue for pathologies, such as endometriosis or endometriosis-associated ovarian cancer [4]. Therefore, its comprehensive analysis, including digital methods, is an urgent and vital task. One of the methods to evaluate these reactions is to calculate a histological [5,6] score—a cumulative quantity representing the receptivity of a sample to a given hormone. Calculating such quantities requires the painstaking calculation of the nuclei of different staining responses. When done manually, this calculation is prone to errors, as only a limited amount of the fields of view are analyzed, and the subjectiveness of an expert is always present. The automatization of such tasks may solve these problems and improve the speed and availability of such analysis.

To create models capable of calculating histological scores, rich, representative, and well-labeled data sets are required. The first step for computing the histological scores is detecting and counting nuclei. The corresponding dataset should have information about their positions and stoma/epithelium classes for various patches taken from histological slides.

Here we present EndoNuke, to our knowledge, the first open dataset dedicated to automating the scoring process of endometrium immunohistochemistry (IHC) slides. We also perform an expert agreement study, which is essential not only for developing detection models but also for estimating the representability of the histological scores themselves. Finally, we perform a simple analysis of the staining properties of the labeled nuclei.

The dataset is available at endonuke.ispras.ru (accessed on 20 April 2022) under the Creative Commons Attribution 4.0 International license [7]. All algorithmic methods are supported with the code, freely available at github.com/ispras/endometrium-dataset-analysis/ (accessed on 20 April 2022).

## 2. Related Work

Currently, there are several histological datasets freely available on the web with significantly diverse purposes, properties, and sizes. In this work, we do not aim to cover all the available data, instead focusing on several of the most representative works in the field.

As staining with hematoxylin and eosin (H&E) is the most common technique in modern pathology, there is no surprise that the datasets based on this protocol are the most numerous and diverse. These datasets set up a base for open competition on platforms, like Kaggle and Grand Challenge.

The dataset with *H&E* slides [8] is for predicting endometrial cancer subtypes and molecular features which is available at The Cancer Imaging Archive (TCIA). This dataset is the most similar to our dataset, and we will compare our results with it.

The PANDA dataset [9] is dedicated to an estimation of the severity of prostate cancer by computing Gleason scores [10]. This is one of the most extensive histology datasets, consisting of more than 10,000 whole slide images (WSI), with roughly half of them having benign/tissue masks and the other half with gland masks.

The Breast histopathology Invasive Ductal Carcinoma (IDC) dataset [11,12] is another dataset dedicated to cancer. It consists of 277,524 small tiles, taken from the slides of breast biopsy, with binary classification labels ($IDC-/IDC+$).

Another breast cancer dataset, BACH [13,14], utilizes four class labeling, with the labels being normal, benign, in situ carcinoma, or invasive carcinoma. The dataset consists of 400 tissue tiles with imagewise labels and 10 WSI with mask labels.

The CAMELYON [15] dataset, created for the detection of cancer metastases in histological slides, consists of 1399 WSI, of which 1190 slides have slide level annotations and 209 slides have mask annotations. The class labels correspond to different metastases types: no metastases, macro-metastases, micro-metastases, and isolated tumor cells.

MoNuSeg dataset [16] labeling is on a lower level than the previous ones. The labels in MoNuSeg are nuclei masks, manually segmented on the $1000 \times 1000$ images taken from 30 WSI (one image per slide) with 22,000 nuclei being detected in total. This dataset is not focused on a single organ, providing the slides from samples taken from the breast, liver, kidney, prostate, bladder, colon, and stomach. Although the total number of images is relatively small, this dataset can be used for detection purposes, as in the detection problem, an individual sample is the single nuclei, not the whole image. This makes this a valuable dataset for low-level tasks.

The Pannuke dataset [17] is distinct from the other datasets described here, as it was generated in a semi-automated manner. Multiple models were trained on several open datasets and used to label new data. After the uncertainty of the predictions was estimated, the most uncertain samples were manually relabelled, and the models were trained again. After several rounds of this procedure, the final dataset was obtained, with different nuclei types as class labels of masked instances on 455 fields of view (FOV).

Another approach to gathering relatively large datasets is crowdsourcing. It was demonstrated in BCSS [18] and NuCLS [19] datasets, which were labeled by groups of experts with diverse levels of expertise. The BCSS dataset was labeled by 25 pathologists, from medical students to seniors, and resulted in more than 20,000 annotated tissue regions with 20 class labels. The NuCLS [18] dataset was annotated by 3 senior pathologists, 4 junior pathologists, and 25 non-pathologists; it consists of 3944 FOV with more than 220,000 labels, including overlapped masks in multi-expert labeling. The involvement of multiple experts with such differing experience levels can significantly benefit the size of the dataset, but requires a thoughtful study of rater concordance and the validation of their work by experienced pathologists. Both datasets [18,19] are supplemented with such a study.

The datasets of images or slides with IHC staining are much less numerous. This includes the Gastrit cancer automatic detection dataset [20], which consists of 7394 nuclei size image patches with binary classification labels. Although the FOV from which the patches were extracted are not available, it is still possible to train a detection model based on this data.

Another interesting example of different staining data is the ANHIR [21] dataset. It consists of 46 sets of images taken from the adjacent slides and stained with different techniques (including IHC and H&E). The images are annotated with landmarks placed on the same physical spots, so it is possible to create spatial mapping between the pairs of images.

## 3. Materials and Methods

### 3.1. Sample Collection

The Commission of Biomedical Ethics at the National Medical Research Center for Obstetrics, Gynecology, and Perinatology and the Ministry of Healthcare of the Russian Federation approved that study was performed according to Good Clinical Practice guidelines and Declaration of Helsinki (Ethic's committee approval Protocol No. 5, 27 May 2021). Each participant provided informed consent for the purposes of the study. Written consent for publication was obtained from the patient or their relative. Inclusion criteria were reproductive age and the presence of an ovarian endometrial cysts, confirmed morphologically after cystectomy. All included patients had secondary infertility, decreased ovarian reserve, and unilateral or bilateral ovarian endometrial cysts. The 12 patients (from 26 to 40 years old) included in the study underwent an aspiration endometrial pipelle biopsy in the middle stage of the proliferation and secretion phases. The biopsy was placed in 10% formalin for fixation.

*3.2. Slide Preparation*

Formalin-fixed pipelle endometrial biopsy underwent routine histological processing using the automatic histoprocessor Leica ASP30 and the station for embedding in paraffin Leica EG1150. A series of endometrial biopsy sections (4 μm) for visualization of general tissue structure and for immunohistochemical analysis were obtained from each patient. Sections were placed in polylysine glass slides and were dried in a drying chamber at 70 °C for at least 2 h. For immunohistochemical analysis, paraffin sections were deparaffinized in a series of containers with xylene, passed through alcohols of descending strength, followed by water and the appropriate buffer solution. Then, antigen unmasking and blocking procedures were performed and sections were transferred to APK Wash buffer and stained in immunohistostainer Ventana BenchMark Ultra (Ventana Medical-Systems, Inc., 1910 E Innovation Park Dr, Oro Valley, Arizona 85755, USA) using antibodies: Ventana CONFIRM anti-Progesterone Receptor (PR) Rabbit Monoclonal Primary Antibody (1E2, cat.num.790-4324, Oro Valley, Arizona, USA), PR-A and PR-B, and Ventana CONFIRM anti-Estrogen Receptor Rabbit Monoclonal Primary Antibody (SP1, cat.num. 790-4324, Oro Valley, Arizona, USA) and immunoperoxidase-conjugated secondary antibodies in dilutions recommended by the manufacturers. Labeling was visualized by reaction with DAB (diaminobenzidine tetrahydro-chloride), and counterstained with hematoxylin. Each section was scanned with a Leica Aperio AT2.

*3.3. Tiles Sampling and Preparation*

After the slides were prepared, we sampled tiles of fixed physical sizes of $100 \times 100$ μm. Due to the differing magnifications on the scanner, the pixel sizes of the images are also different: $200 \times 200$ pixels for 0.5 μm/pixel resolution and $400 \times 400$ pixels for 0.25 μm/pixel resolution. The tiles are stored in their original sizes, however, for the research presented in this paper, we rescaled them to a unified size of $256 \times 256$.

The sampling process was the following: first, a grid was created on the slide, then the tiles were uniformly sampled from the grid. A simple script detected whether the tile was predominantly a background one—these tiles were excluded from the resulting set.

As many tiles did not contain any information about the nuclei, we decided to manually filter some of the tiles before labeling. As this process can skew the distribution of the images, we filtered only one third of the dataset. The filtered and unfiltered parts of the dataset can be easily restored from the file IDs (see instructions on the dataset website).

Each tile is supplemented with a context image (see Figure 1)—the image of a physical size of $300 \times 300$ μm, with the tile of interest put into the contrast frame. This context is not assumed to be labeled, as it serves only to provide the information about the larger tissue structure.

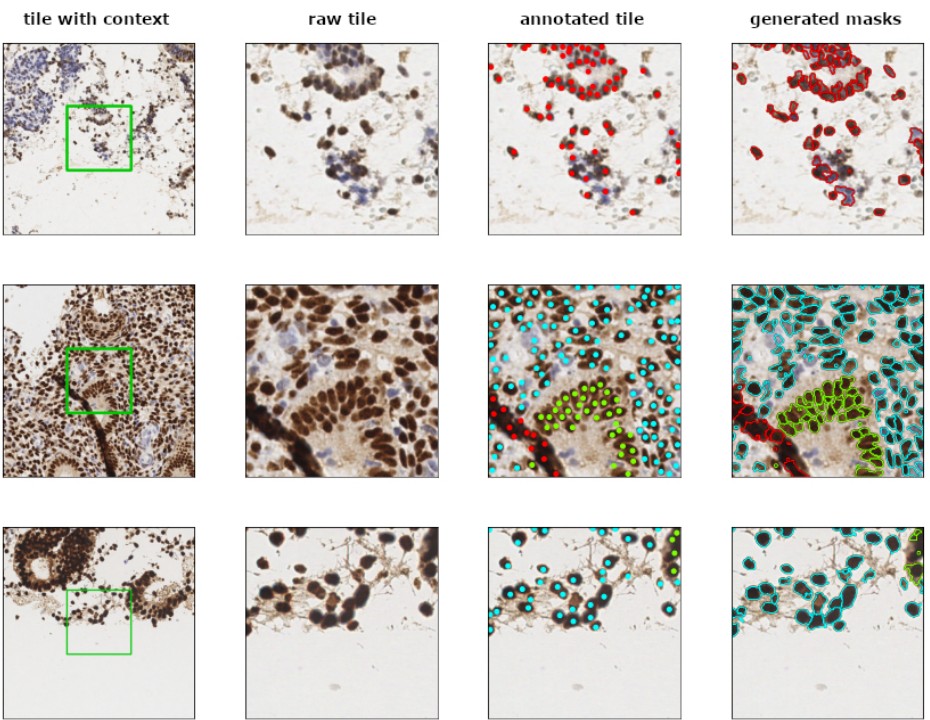

**Figure 1.** Context image, raw tile, labeled tile, and a tile with masks. Cyan, green, and red labels correspond to "stroma", "epithelium", and "other" classes, respectively. "Raw tile" is an image from the dataset without annotation, "tile with context" is a tile with the surrounding areas , "annotated tile" is a tile with the nuclei annotated by the experts, "generated masks" is a tile with the masks generated by watershed algorithm.

### 3.4. Annotation Protocol

The annotation was conducted in three phases with the help of the CVAT web application tool [22]. For labels, the "points" tool was used—although it gives only information about nuclei location, it reduces the annotation time, as the expert does not have to draw a bounding box or a mask.

In the first phase, all experts were gathered together in a virtual room, and a briefing from a senior pathologists was provided. After a Q&A session, a fixed set of 10 tiles was labeled by each expert. All seven experts could not communicate with each other during this process. After the labeling was completed, an additional Q&A session was conducted. The annotation gathered in this phase was used in what we call the "preliminary" agreement study.

In the second phase, the bulk of the dataset was created. Each expert annotated their own tiles, grouped in tasks of 20 tiles. As a task was finished, a new task was provided. The experts were able to communicate with each other via a chatroom. There was an additional set of 10 tiles, which were mixed among the first few tasks of each excerpt. These 10 tiles were used to estimate agreement after the second Q&A session in phase 1 and construct the "hidden" agreement study.

The third phase was somewhat similar to the first phase. The experts were given 20 tiles each, but there was no briefing or Q&A session. They were also not informed about this study. This study was performed to estimate the agreement after all raters got some experience and communication was almost absent. These tiles are the basis for the "posterior" agreement study.

After the third phase, the experts continued to annotate the dataset according to the protocol of phase 2. In total, the experts annotated the dataset over one month. The CVAT web application was hosted on the ISPRAS server and was available during

the entire annotation period, which allowed the experts to access the data at any time they found convenient.

### 3.5. Generation of the Masks

Although our annotation does not have any information about nuclei shapes, it is possible to restore them using classical computer vision algorithms. To generate these masks we utilise a watershed [23] algorithm from the skiimage library [24]; around each nucleus, a window of fixed size is created, then all keypoints in the scope of the window are used as seed points. To find the seeds for the background, we use the MultiOtsu method [25] with three classes.

In most cases, this method allows us to accurately determine the borders of the nuclei. However, if a nucleus almost merged with another object (i.e., another nucleus) or the nucleus boundary is fuzzy, the algorithm fails to correctly determine its shape. As the misshapen masks usually have areas that are too large or too small, we treated them as outliers and found them by determining two thresholds: for the small and large nuclei.

To define these thresholds we used the following methodology. First, we aggregated the raw distribution without any thresholds, leaving all masks as they were generated by the watershed algorithm (Figure 2, left). We observed a smooth peak in the middle of the distribution and a sharp peak on the left corresponding to outliers with small outliers. By estimating a probability density using a Gaussian kernel, we found a center of the well between the peaks and treated it as a threshold for small outliers. To find the threshold for large outliers, we found the right 99th-percentile (starting from the center of a smooth peak).

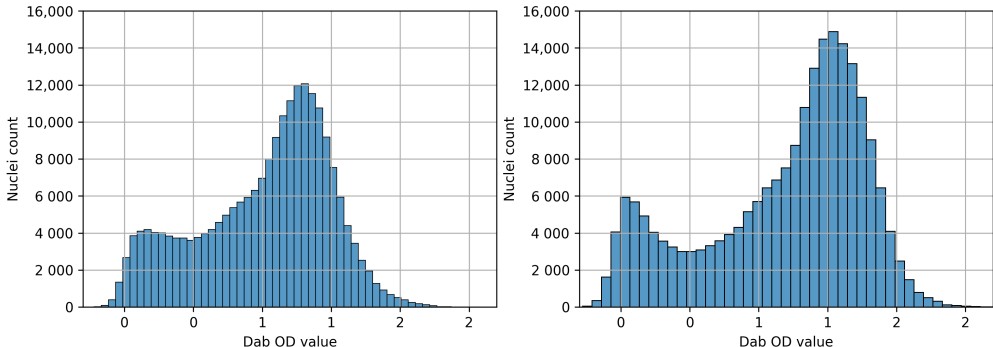

**Figure 2.** The empirical nuclei mask size distribution. The left distribution is based on the unfiltered masks that still contain the nuclei with radii that were too small or large and were excluded from the primary dataset, as they were considered to be artifacts of the algorithm. The right distribution is the filtered data, which is the basis for the typical radius estimation.

### 3.6. Expression Rate Analysis

After the masks were generated, it was possible to calculate the expression rate of a nucleus. We used a basic technique of color deconvolution to extract the DAB optical densities (OD), as described in [26]. For all pixels inside the mask, the DAB OD values were calculated and the mean value was taken.

As our masks are not perfect, we also tried another approach to estimate the expression rate, which we call a probe approach. Instead of generating the masks with a watershed algorithm, we made a circular mask around each keypoint with a radius smaller than an average nuclei radius. This allowed us to "take" only a limited amount of pixels into a "probe", which, generally, are not close to the boundary of the nucleus and, thus, may better represent the staining. The masks which we generated by the probe method have a physical radius of 1 μm.

### 3.7. Agreement Study

The first step in studying agreement is to define the objects of interest. In our case, the objects are nuclei, so we had to treat them as individual samples. This implies two problems: the first is that the nuclei, found by different experts on the same tiles, are not at the exact same positions, and the second is that a given nuclei can be labeled by one expert and not labeled by another. We address these problems using the following methodology.

For each pair of experts, we needed to create a matching between the keypoints. To do so, we choose a similarity measure as keypoint similarity from [27]:

$$KS(x_i, x_j) = \exp\left(-\frac{|x_i - x_j|^2}{s^2}\right) \tag{1}$$

Here $x_i$, $x_j$ are point locations and $s$ is a scale parameter, which represents the typical nuclei size. We chose this parameter equal to the mean square radius, which we are able to estimate using the generated masks. Now we compute the similarities $KS(x_i, \tilde{x}_j)$ for all pairs of $x_i$, $\tilde{x}_j$, where $x_i$ is the location of the point from the first annotation and $\tilde{x}_j$ is the location of the point from the second annotation. These similarities form a matrix, which can be interpreted as an adjacency matrix for a bipartite graph. We applied the Hungarian algorithm [28] to solve the linear assignment problem of this matrix, which allowed us to find a matching between the keypoints from two annotations with maximum total similarity.

If the number of points differs, some points will not be used in the matching. There are two strategies to deal with them. The first is to simply drop these points. This approach allows us to estimate the agreement on keypoint class labels, but not on the keypoint numbers. The second approach is to give all the missed points an extra class ("background") and measure the agreement statistics. This approach reduces the observed concordance metrics, but allows us to estimate both misscalsifications and the missing points in a single metric. For all agreement studies (preliminary, hidden, posterior) we have used both approaches, as both of them provide meaningful information about annotation quality.

After the matching was performed, we computed the Kohen's kappa [29] statistic for each pair of experts. As advised in [30,31], we used mean kappa to obtain the final agreement measure of the cohort. For each study we measured three mean kappas—the first was averaged over the pathologist-pathologist pairs, the second was averaged over the student-student pairs, and the third was averaged over pathologist-student pairs. The first two values provide information about the agreement inside two subcohorts, and the third one shows to which extent less experienced annotators agree with more experienced ones.

## 4. Results

### 4.1. The Annotated Dataset

The resulting dataset consists of two separate parts. The first consists of 40 tiles with 34,701 nuclei keypoints and was labeled by all experts who participated in the study (three pathologists and four medical students). This part was used for the agreement study and can also be utilized as a test dataset with high quality labels obtained by merging the keypoints from the experienced pathologists.

The second part consists of 1740 tiles with 210,419 nuclei labels. Each of these tiles was labeled by a single expert. 268 tiles were labeled by experienced pathologists, while the rest of the dataset was created by medical students.

### 4.2. Mean Nuclei Radius

With the help of the generated masks, we were able to estimate the typical size of a nucleus in our sample. To do so, we obtained the distribution of mask sizes, cut off the small and the large outliers, calculated mean mask size, $S_{mean}$, and estimated the typical

radius as $\sqrt{S_{mean}/\pi}$. The resulting number is 7.63 pixels for a 256 × 256 image size, which approximately corresponds to 2.98 µm. The distribution of the areas is shown on Figure 2.

We also separately calculated typical sizes of stromal and epithelial nuclei and found them to be 2.92 µm and 3.24 µm, respectively. The epithelial component of the endometrium is formed mostly by secretory and ciliated cells, while stroma predominantly consists of fibroblast-like cells which actively synthesizing collagen and glycosaminoglycans. The origin of these cells is different, as well as their morphology and functional activity, which may be the reason for the different size of the nuclei [32]. In addition, nuclei of stromal cells can be compressed by the neighboring myometrium in the basal layers of the endometrium in the junctional zones.

The distribution of nuclei (both stromal and epithelial) DAB OD values obtained by the two methods discussed earlier is presented on Figure 3. By using the dip test [33], we rejected the unimodality hypothesis for both distributions with a *p*-value of $5 \times 10^{-6}$ for the full mask method and with a *p*-value less than $2.2 \times 10^{-16}$ for the probe method.

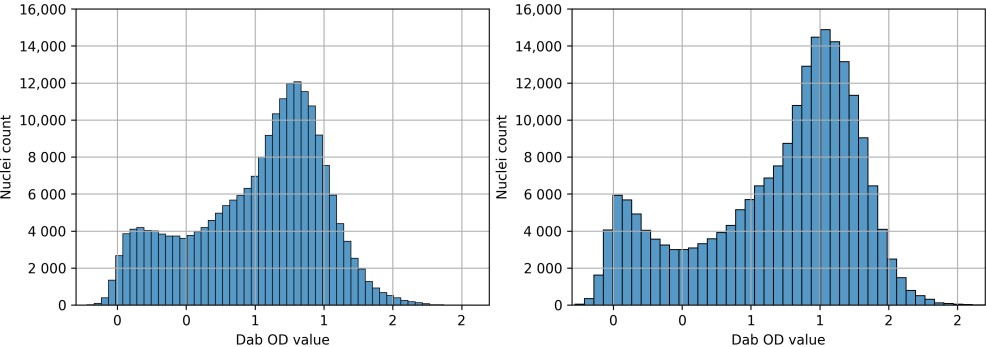

**Figure 3.** Nuclei DAB OD value distribution obtained by two methods: full masks (**left**) and probes (**right**).

### 4.3. Agreement Study

The pairwise Cohen's kappas for each expert pair are presented on Figures 4 and 5 for both strategies—dropping unmatched points and keeping them as background samples.

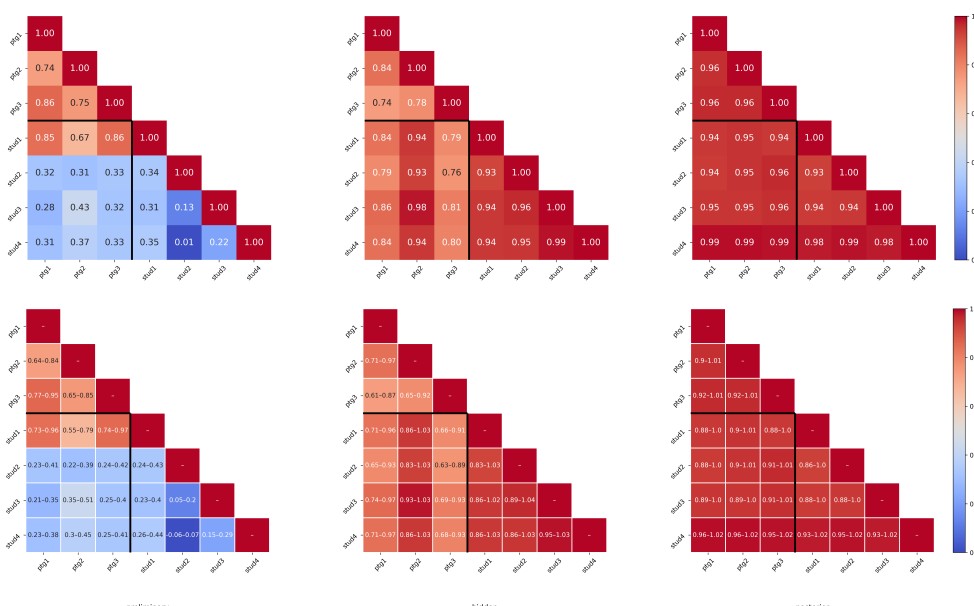

**Figure 4.** Cohen Kappa values between all the annotations and confidence intervals for them with the method where the unmatched keypoints were dropped. Upper row are the kappa values, lower row are the corresponding 95% confidence intervals.

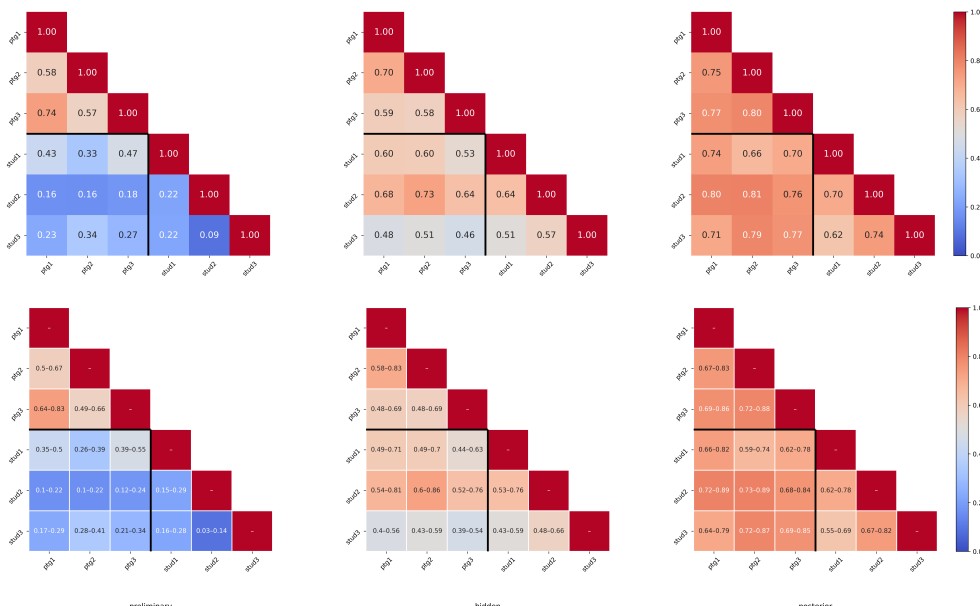

**Figure 5.** Cohen Kappa values between all the annotations and confidence intervals for them with the method where the unmatched keypoints were assigned a separate class ("background"). The upper row is the kappa values, the lower row is the corresponding 95% confidence intervals.

The mean kappas between the cohorts are presented in Table 1

We tend to use mean kappa value for posterior studies as a final measure of expert agreement, as it was measured roughly in the middle of the annotation process. Using kappa for the second strategy (unmatched nuclei as background class) and interpretation as suggested in [34], we interpreted pathologist-pathologist agreement as moderate (mean kappa 0.77), student-student as weak (mean kappa 0.50), and pathologist-student agreement as moderate (mean kappa 0.64).

**Table 1.** Mean Kappas between the groups. Here "preliminary", "hidden" and "posterior" agreement was estimated prior, during the process and after the creation of the bulk of the dataset, respectively. More details in Section 3.4.

| Agreement Study | Preliminary | | | Hidden | | | Posterior | | |
|---|---|---|---|---|---|---|---|---|---|
| | ptg ptg | stud stud | ptg stud | ptg ptg | stud stud | ptg stud | ptg ptg | stud stud | ptg stud |
| Dropped unmatched nuclei | 0.78 | 0.22 | 0.45 | 0.79 | 0.95 | 0.86 | 0.96 | 0.96 | 0.96 |
| Unmatched nuclei as background class | 0.63 | 0.15 | 0.27 | 0.62 | 0.47 | 0.53 | 0.77 | 0.50 | 0.64 |

### 4.4. Annotation Quality

Our primary goal was to create a rich and well-labeled dataset that can be used for creating models for nuclei detection on endometrium slides. As our main benchmark for the annotation quality is expert agreement, it was important to track how it changed during the creation of the dataset. In the preliminary agreement study, the pathologists demonstrated a decent agreement, while the students agreement was poor, both with each other, and with the pathologists. It indicates that the briefing and Q&A session that was conducted before that phase was not sufficient for the creation of a consistent annotation. However, during the preliminary agreement phase, the students accumulated

new questions and attained experience. After the additional Q&A session the agreement noticeably increased—the hidden agreement study showed much larger kappas values than the preliminary agreement study. However, we realize that, as all the experts were able to communicate with each other during this phase, the kappa values may be overestimated. Finally, the posterior agreement study, which was conducted roughly in the middle of the annotation process, shows further improvement of the metrics. At that time, the communication between the experts was not intensive, so the overestimation factor should not be significant. We also note in that phase the agreement between the pathologists is close to the agreement between most of the students and the pathologists, which indicates that the students acquired sufficient experience.

Although the task of annotating the nuclei seems to be relatively simple, the agreement between any pair of experts rarely reaches 0.8 (for the method without dropping missed nuclei), including the pathologist-pathologist pairs.

This indicates that even annotation in such tasks requires a thoughtful agreement study. Moreover, the preliminary agreement study shows the annotations created by inexperienced personnel trained only by briefing will probably be incomplete, and the class labels may often be incorrect.

Moreover, as the agreement between even experienced specialists is not perfect, indicating that histological scores obtained manually are also affected by object interpretation (nuclei/not nuclei) errors. If an end-to-end model is trained on a dataset with histological scores as annotations, it will likely inherit these errors, not to mention the errors caused by the subjectivity of staining interpretation. Thus, we believe that scoring tasks on histological data should not be solved in an end-to-end manner, and object detection models with subsequent object counting should be used, as they provide more interpretable results.

*4.5. The Expression Rate Distribution*

Despite the fact that both distributions presented on Figure 3 are bimodal, we must state that the well in the middle of the distribution may have been caused not by a natural separation of the positive and negative nuclei, but by the imperfections of the staining extraction algorithm. Although our distributions resemble each other, they are not the same— the two-sample Kolmogorov-Smirnov test [35,36] verifies this with *p*-value less than 2.2–16. This indicates that the shape of the distribution is dependent on the method of staining calculation and thus the quantities obtained from them may not be universal.

Moreover, even if there is a well in the middle of the expression rate distribution, only two classes can be defined—that of positively and negatively stained nuclei. To implement methods which separate the nuclei for more than two classes, one has to define several thresholds, so some of them will not correspond to any well in staining distribution. This indicates that the classes separated by these thresholds would be subjective and probably not reliable, so histological scores based on them would be prone to errors. Thus, for scoring the IHC slides stained with estrogen and progesterone, we advise using the scores with two classes of nuclei.

## 5. Conclusions

In this study, we created a dataset for nuclei detection in endometrium IHC slides which consists of 1780 annotated tiles. The agreement study shows that the annotations created by the pathologists are not perfect, but are reliable enough to be used as ground truth labels. Label quality from less experienced specialists is lower, but also varies from poor to moderate, so the question of their usage should be decided by the researcher creating the model. When compared to the *H&E* dataset [8], ours is similar, but has another purpose. Our dataset is for the detection of endometrium cells, but the *H&E* dataset is for their classification.

In addition, analysis of nucleus staining shows the two class scores are preferable for estrogen and progesterone expression rate estimation. The vast majority of automatic labeling tools in biomedicine have been developed for oncomorphology, the most relevant and

popular area of pathological anatomy, where the task of machine learning is to distinguish tumor tissue from healthy tissue [37–39]. Nevertheless, such developments would also be in great demand in the field of assisted reproductive technologies (ART), where automatization of endometrium labeling can significantly improve the outcomes of programs. Such an approach could be used as a diagnostic tool for analyzing endometrium receptivity and determining the implantation window. Our annotated dataset a step that brings us closer to spreading this approach in ART protocols.

**Author Contributions:** Conceptualization, T.F. and A.N.; methodology, A.N., T.K. and A.K.; software, A.N., A.I. and E.U.; validation, A.N. and T.F.; formal analysis, A.N.; investigation, A.N. and E.U.; resources, K.M. and S.N.; data curation, K.M. and S.N.; writing—original draft preparation, A.N., P.V. and E.U.; writing—review and editing, A.N. and E.K.; visualization, A.N.; supervision, L.M.; project administration, T.F.; funding acquisition, E.K. and T.F. All authors have read and agreed to the published version of the manuscript.

**Funding:** A part of the work concerning tissue preparation was supported by the Ministry of Health of Russian Federation within the framework of State Assignment No. 121040600436-7. A part of the work concerning the detection of nuclei was supported by the Ministry of Science and Higher Education of the Russian Federation, agreement No. 075-15-2022-294 dated 15 April 2022.

**Institutional Review Board Statement:** The Commission of Biomedical Ethics at the National Medical Research Center for Obstetrics, Gynecology, and Perinatology and the Ministry of Healthcare of the Russian Federation approved that study was performed according to Good Clinical Practice guidelines and Declaration of Helsinki (Ethic's committee approval Protocol No. 5, 27 May 2021). Each participant provided informed consent for the purposes of the study. Written consent for publication was obtained from the patient or their relative.

**Informed Consent Statement:** Informed consent was obtained from all subjects involved in the study.

**Data Availability Statement:** http://endonuke.ispras.ru/ (accessed on 20 April 2022).

**Conflicts of Interest:** The authors declare no conflict of interest.

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
