# Peer review of "EndoNuke: Nuclei Detection Dataset for Estrogen and Progesterone Stained IHC Endometrium Scans"

_data, 2022_

Round 1

Reviewer 1 Report

I read with great interest the manuscript, which falls within the aim of this Journal. In my honest opinion, the topic is interesting enough to attract the readers’ attention. Nevertheless, authors should clarify some points and improve the discussion, as suggested below.

Authors should consider the following recommendations:

  • Manuscript should be further revised in order to correct some typos and improve style.
  • I would suggest to discuss, at least briefly, the suitability of eutopic endometrium as control for endometriosis research, underlining strong and weak point about the field (authors may refer to: PMID: 35454018; PMID: 35069072).

Author Response

Discuss about the suitability of eutopic endometrium as control for endometriosis research, underlining strong and weak point about the field (authors may refer to: PMID: 35454018; PMID: 35069072).

Dear Reviewer,

Thank you for this comment. We added the following text in the Introduction section: “The endometrium is not only the subject of histopathological diagnosis but also an extremely valuable tissue for experimental manipulations and modeling of reproductive diseases using human biomaterial. For instance, the endometrium could be used as a control tissue for pathologies such as endometriosis or endometriosis-associated ovarian cancer. Therefore, its comprehensive analysis, including the use of digital methods, is an urgent and important task.” on 2 page.

Reviewer 2 Report

In the manuscript presented open dataset dedicated to automating the scoring process of endometrium immunohistochemistry slides. The dataset has been annotated by several experts with various experiences. This dataset could be used for creating software tools that play a supporting role in the process of making a medical or clinical decision

The study included 28 patients, but only 10 estrogen receptor and 9 progesterone receptor slides were used for further analysis. Which criteria were used to exclude some samples?

This article does not have a list of references, so it is difficult to assess whether the authors refer to new relevant studies. In some places, references are given to methods and algorithms, but since there are no citation sources, it is impossible to evaluate them.

Table 2 shows Mean Kappas, for example, Dropped unmatched nuclei ptg/ptg is 0.96, but in figure 4, the maximum value of Kappa is 0.80, how did authors get such an average?

The materials and methods section is good and detailed, it also contains the interpretation of the results, while the results section consists of only 4 lines. It is better to transfer some of the information from the materials and methods to the results section.

In the second part of annotated dataset 268 tiles were labeled by experienced pathologists, the rest of the dataset was created by medical students. Why did unqualified personnel labelled most of the tiles and did this affect the accuracy of the results?

Author Response

Dear Reviewer,

Thank you for appreciating our article. In the revised version of the manuscript, we added several text fragments and tried to resolve all issues:

  • The study included 28 patients, but only 10 estrogen receptor and 9 progesterone receptor slides were used for further analysis. Which criteria were used to exclude some samples?

Indeed, after careful analysis, we left only 19 slides from the entire sample, which correspond to 12 patients. This is explained by heterogeneity of patients and the need to exclude some patients after analyzing clinical and anamnestic characteristics. We added more patient`s data and corrected the original number of participants (n=12) included in the analysis on page 3: “Inclusion criteria were reproductive age and the presence of an ovarian endometrial cysts, confirmed morphologically after cystectomy. All included patients had secondary infertility, decreased ovarian reserve, and unilateral or bilateral ovarian endometrial cysts. The 12 patients (from 26 to 40 years old) included in the study underwent an aspiration endometrial pipelle biopsy in the middle stage of the proliferation and secretion phases”.The number of 19 reflects that several patients had slides for both progesterone and estrogen receptor staining. We added this information in the metadata of the slides.

  • This article does not have a list of references, so it is difficult to assess whether the authors refer to new relevant studies. In some places, references are given to methods and algorithms, but since there are no citation sources, it is impossible to evaluate them.

This oversight happened by accident. A list of references certainly exists; please find it in the revised version of the Manuscript. More than 65% of the references in it are articles published after 2015.

  • Table 2 shows Mean Kappas, for example, Dropped unmatched nuclei ptg/ptg is 0.96, but in figure 4, the maximum value of Kappa is 0.80, how did authors get such an average?

Thank you for noticing this. We have uploaded the right figures in the manuscript on page 9.

  • The materials and methods section is good and detailed, it also contains the interpretation of the results, while the results section consists of only 4 lines. It is better to transfer some of the information from the materials and methods to the results section.

We have transferred some information to the Results section on page 11.

  • In the second part of annotated dataset 268 tiles were labeled by experienced pathologists, the rest of the dataset was created by medical students. Why did unqualified personnel labelled most of the tiles and did this affect the accuracy of the results?

Indeed, most of the tiles were annotated by students. This was due to the fact that qualified doctors have little time for annotation, it is cheaper to hire students than qualified doctors, due to their number, they annotate faster and students are easily taught this task (agreement can be seen in Figures 4 and 5).

Thank you for these comments. Hope you find the changes satisfied.

Reviewer 3 Report

Comments to the author:

IHC analyses traditionally have been semi-quantified by pathologists' visual scoring of staining. IHC is useful for validating biomarkers discovered through genomics methods as large clinical repositories. However, pathologists' visual scoring is sometimes fraught with problems due to subjectivity in interpretation. For this article, “EndoNuke: nuclei detection dataset for estrogen and progesterone stained IHC endometrium scans”, the reviewer has some concerns and criticisms that should be improved, which are listed below:

  1. Abstract: please explain “...analyzed the staining properties of the nuclei” in more detail!
  2. Sample collection: please explain the characteristics patient in this research, ages, healthy/unhealthy endometrium, the processing sample collection into paraffin section and how many samples in total from 28 patients can be produced?
  3. Please mentions the detailed characteristic of the sample used in this research (the thickness of the paraffin cut, the angulation of the sectional samples) on the materials and methods.
  4. How do the author distinguish the nucleus of estrogen-positiveor progesterone-positive or double estrogen-progesterone-positive? What are the antigen-antibodies that were used in this sample?
  5. Figure 1, please add a title for each figure and explain more is it taken from manual or automated annotation, what is the difference (from the left, the 3rd and 4th figures dot and area)? 
  6. Please describe the summary of the questions and answers (Q&A) that were used by the pathologist and student for every phase.
  7. Please add discussion comparing endoNuke result to other open datasheet (example: endometrium related infertility or other pathogenesis condition)
  8. Please explain in more detail the conclusion of this studyand the novelty and benefit of this study for human fertilization study, as mentioned in the introduction.
  9. Is there any difference in labelling, scoring, detecting the nucleus from the glands and stroma of endometrium? Is there any different in the size of the nucleus from this part (glands and stroma)? please explain on the result/ discussion.
  10. Figure 4, Table 1 please explain the preliminary, hidden, and posterior means on the figure legend. In addition, please add confidence intervals for Kappa analysis.
  11. Please add the cohen's kappa statistical significance or agreement range score for figure 4 and table 1 interpretation.
  12. Materials and methods; 3.8 the annotated data sheet:”the first consist of 40 tiles with 34701 nuclei keypoints and was labeled by all experts who participated in the study.” and “The second part consists of 1740 tiles with 210419 nuclei labels. Each of these tiles was labeled by a single expert.” while in the result: “We successfully annotated more than 1600 tiles sampled with nuclei locations and tissue labels.” Please explain the result the process and the elimination to get more than 1600 tiles samples.
  13. Result: Please explain “The analysis of staining distribution implies that only two classes of staining intensities are well defined in these samples.” in more detail!
  14. The result and/or discussion can be written better if the author limited the information on the materials and methods.

Author Response

Dear Reviewer,

Thank you for appreciating our article. In the revised version of the manuscript, we added several text fragments and tried to resolve all issues:

  1. Abstract: please explain “...analyzed the staining properties of the nuclei” in more detail!We have changed “analysed the staining properties of the nuclei” to “analysed the distribution of nuclei staining intensity”.
  2. Sample collection: please explain the characteristics patient in this research, ages, healthy/unhealthy endometrium, the processing sample collection into paraffin section and how many samples in total from 28 patients can be produced?Thank you for this note. After careful analysis, we left only 19 slides from the entire sample, which correspond to 12 patients. This is explained by heterogeneity of patients and the need to exclude some patients after analyzing clinical and anamnestic characteristics. We added more patient`s data and corrected the original number of participants (n=12) included in the analysis on page 3: “Inclusion criteria were reproductive age and the presence of an ovarian endometrial cysts, confirmed morphologically after cystectomy. All included patients had secondary infertility, decreased ovarian reserve, and unilateral or bilateral ovarian endometrial cysts. The 12 patients (from 26 to 40 years old) included in the study underwent an aspiration endometrial pipelle biopsy in the middle stage of the proliferation and secretion phases” Also we added the following text: “A series of endometrial biopsy sections (4 μm) for visualisation of general tissue structure and for immunohistochemical analysis were obtained from each patient.” The number of 19 slides reflects that several patients had slides for both progesterone and estrogen receptor staining. We added this information in the metadata of the slides.
  3. Please mentions the detailed characteristic of the sample used in this research (the thickness of the paraffin cut, the angulation of the sectional samples) on the materials and methods.Now we added more details in this section. Thank you for this suggestion.
  4. How do the author distinguish the nucleus of estrogen-positiveor progesterone-positive or double estrogen-progesterone-positive? What are the antigen-antibodies that were used in this sample?We haven't used the approach with double-positive staining. We use estrogen and progesterone receptors staining separately.  Please, find the information about antibodies on page 4: “antibodies: Ventana CONFIRM anti-Progesterone Receptor (PR) Rabbit Monoclonal Primary Antibody (1E2, cat.num.790-4324, USA), PR-A and PR-B, and Ventana CONFIRM anti-Estrogen Receptor  Rabbit Monoclonal Primary Antibody (SP1, cat.num. 790-4324, USA) and immunoperoxidase-conjugated secondary antibodies in dilutions recommended by manufacturers”.
  5. Figure 1, please add a title for each figure and explain more is it taken from manual or automated annotation, what is the difference (from the left, the 3rd and 4th figures dot and area)? The first image (from left to right) is a tile with a context, the second is a raw tile, the third is annotated by experts tile, the fourth is a tile with automatically (by algorithms) generated masks. We have added explanations of differences to the capture.
  6. Please describe the summary of the questions and answers (Q&A) that were used by the pathologist and student for every phase.Now we do not have a full annotation protocol for students.  But the most popular question was about controversial nuclei on tiles. Example of this tile you can see in attachment.
  7. Please add discussion comparing endoNuke result to other open datasheet (example: endometrium related infertility or other pathogenesis condition).We have added comparison with most similar dataset to the Conclusion section.
  8. Please explain in more detail the conclusion of this studyand the novelty and benefit of this study for human fertilization study, as mentioned in the introduction.Thank you for this recommendation. We added a text in the Conclusion section.
  9. Is there any difference in labelling, scoring, detecting the nucleus from the glands and stroma of endometrium? Is there any different in the size of the nucleus from this part (glands and stroma)? please explain on the result/ discussion.Thank you for this note. We also separately calculated typical sizes of stromal and epithelial nuclei and found them to be 2.92 mcm and 3.24 mcm respectively. We have added information about that on page 8.
  10. Figure 4, Table 1 please explain the preliminary, hidden, and posterior means on the figure legend. In addition, please add confidence intervals for Kappa analysis.We have added confidence intervals for kappas and explanations to the captions. You can see it on Figures 4 and 5.
  11. Please add the cohen's kappa statistical significance or agreement range score for figure 4 and table 1 interpretation.We have used Table 3 from https://www.ncbi.nlm.nih.gov/pmc/articles/PMC3900052/ to the kappa interpretations which are cited in the text(34).
  12. Materials and methods; 3.8 the annotated data sheet:”the first consist of 40 tiles with 34701 nuclei keypoints and was labeled by all experts who participated in the study.” and “The second part consists of 1740 tiles with 210419 nuclei labels. Each of these tiles was labeled by a single expert.” while in the result: “We successfully annotated more than 1600 tiles sampled with nuclei locations and tissue labels.” Please explain the result the process and the elimination to get more than 1600 tiles samples.We poorly expressed in this section. We have annotated 1780 slides in total.
  13. Result: Please explain “The analysis of staining distribution implies that only two classes of staining intensities are well defined in these samples.” in more detail!The analysis of staining distribution implies that only two classes of staining intensities are well defined in these samples because the distribution of intensities of stains are bimodal as seen from Figure 3.
  14. The result and/or discussion can be written better if the author limited the information on the materials and methods.We have transferred some information to the Results section.

Thank you for these comments. Hope you find the changes satisfied.

Round 2

Reviewer 3 Report

the authors have responded all the reviewer comment, and it is all accepted in the present form